# Construct Validity and Reliability of the Work Environment Assessment Instrument WE-10

**DOI:** 10.3390/ijerph17207364

**Published:** 2020-10-09

**Authors:** Rudy de Barros Ahrens, Luciana da Silva Lirani, Antonio Carlos de Francisco

**Affiliations:** 1Department of Business, Faculty Sagrada Família (FASF), Ponta Grossa, PR 84010-760, Brazil; 2Department of Health Sciences Center, State University Northern of Paraná (UENP), Jacarezinho, PR 86400-000, Brazil; luciana.lirani@uenp.edu.br; 3Department of Industrial Engineering and Post-Graduation in Production Engineering, Federal University of Technology—Paraná (UTFPR), Ponta Grossa, PR 84017-220, Brazil; acfrancisco@utfpr.edu.br

**Keywords:** work environment, research instrument, quality of life, quality of work life, organizational climate

## Abstract

The purpose of this study was to validate the construct and reliability of an instrument to assess the work environment as a single tool based on quality of life (QL), quality of work life (QWL), and organizational climate (OC). The methodology tested the construct validity through Exploratory Factor Analysis (EFA) and reliability through Cronbach’s alpha. The EFA returned a Kaiser–Meyer–Olkin (KMO) value of 0.917; which demonstrated that the data were adequate for the factor analysis; and a significant Bartlett’s test of sphericity (*χ²* = 7465.349; *Df* = 1225; *p* ≤ 0.000). After the EFA; the varimax rotation method was employed for a factor through commonality analysis; reducing the 14 initial factors to 10. Only question 30 presented commonality lower than 0.5; and the other questions returned values higher than 0.5 in the commonality analysis. Regarding the reliability of the instrument; all of the questions presented reliability as the values varied between 0.953 and 0.956. Thus; the instrument demonstrated construct validity and reliability

## 1. Introduction

A dynamic work environment marked by fast and continuous changes, both in social relations and work transformations, demands that managers and companies rethink their positions regarding the workplace, work methods, and the elements that compromise the realization of work activities by the worker adequately [1]. The work environment is explained by the atmosphere of an organization in which the employees perform their labor activities [2]. Work represents a human action performed in a social context and, as such, it is influenced by several sources, which result in a reciprocal action between the collaborator and the work environment [3]. The workplace also contains the interaction between people, work, family, and life [4]. Therefore, productivity requires an appreciation for the environment in which the workers perform their activities. Employees spend a considerable part of their lives working, and a substandard environment entails costs for companies and individuals [5]. Greater productivity requires that companies provide a suitable workplace for that.

Understanding the complex work environment is quite valuable for organizations, as it enables them to elucidate situations that generate health problems and consequently entail high absence rates and turnover, and low productivity at work. In this context, managers’ responsibility for the health and welfare of the workers has increased [6,7]. Environmental assessment is one of the strategic tools that analyzes the internal and external factors of the company [8] and must be employed to find methods to protect workers’ health and integrity. Thus, it is possible to stand out in the market among the competition, achieving business success.

Health-focused environmental assessment is conducted through research instruments or tools, investigating situations that might compromise workers’ health and consequently affect productivity at the workplace. Assessing this environment is a complex task, seeing that it comprises both micro-environmental (internal and external) and macro-environmental (external) factors, which influence an organization’s progress [9].

These factors in the work environment include quality of life (QL), quality of work life (QWL), and organizational climate (OC). Health condition is a fundamental aspect of QL and work capability [10], and a person’s health status is directly influenced and shaped by their work [11]. Work capability is assessed by the individual’s self-perception on their health, work, and lifestyle, comprising physical, mental, and social preconditions [12] and involving the balance between work and personal life.

Similarly, QWL is seen as the degree in which an individual satisfies their personal and work needs through participation at the workplace [13]. It is related to work satisfaction, productivity, health, work safety, motivation, and welfare [14,15].

Finally, OC is linked to motivation, loyalty, and the identity of each company [16]. As collaborators seek recognition based on the quality of the work environment, they share knowledge and attempt to innovate within organizations [17]. It is considered as a significant element, seeing that motivated employees result in more passion for the business, and more engagement with the customers entails higher productivity [18].

It is noteworthy that the human being is complex and holistic, and therefore seeks in an individualized way to satisfy his needs. The theoretical concept of the three focus areas of this study, namely, QL, QWL, and OC, have the same strand and these focus areas are directly related to welfare, satisfaction in the workplace, and productivity, and when used in a unique way they become useful tools for managerial practice.

In the current research scenario, work environment assessment tools or instruments focus on three areas: QL, QWL, and OC [19]. They feature individual applicability, each with its specific identification component and, consequently, individual results.

In this context, the applicability of these individual instruments becomes costly and challenging, as the investigation takes longer and the results cannot be compared and linked to other tools, thus returning unreliable responses. Therefore, the scientific gap for the development of this study reveals itself, with the realization that companies need practical and all-encompassing tools that provide more complete responses about the work environment.

Taking that into account, this study’s general purpose was to validate the construct and reliability of an instrument capable of assessing the work environment as a single tool based on QL, QWL, and OC.

This study is justified because, despite the existence of QL and QWL assessment instruments, their structures are based on classic theoretical models developed over two decades ago, specifically for the American labor culture [20]. Similarly, OC assessment instruments were created in the 1970s, also based on the American line of thought.

The proposal of this new instrument model intends to fill this gap in the academic and organizational context. It will allow the researcher or manager to identify, with a single tool, the problems and/or situations in the work environment that might potentially compromise workers’ health and affect productivity, be it related to QL, QWL, or OC.

## 2. Materials and Methods

### 2.1. Theoretical Framework of the Construct

The first element in the development of a research instrument is the construction of a conceptual structure (theoretical construct) containing the questions and factors related to the theme [21]. Observing the classic QL, QWL, and OC models and following the molds of the instruments WHOQOL [22], Timossi [23], Sbragia [24], and Bispo [25], the domains and criteria of each instrument were identified and compared, listing their similarities. Then, the theoretical construct of the instrument was created with the factors and questions.

The theoretical construct initially contained 23 factors and 83 questions. Afterward, a board of examiners composed of five experts (specialists in psychometric assessment of instruments and work in the areas covered by the instruments) in the field assessed the instrument and suggested the removal of 9 factors and 23 questions, resulting in a final theoretical construct comprising 14 factors and 50 questions. Table 1 displays the domains of the theoretical construct.

The factors represented the areas encompassed by the instruments (QL, QWL, and OC). Within them, factors represented by questions were created, aiming to assess the collaborator’s satisfaction.

### 2.2. Instrument Validation

After establishing the instrument’s theoretical construct, its validation was started. The validity of an instrument is directly related to how it measures what it intends to measure [27]. Thus, an instrument is valid when its construction and applicability allow it to measure its target.

The validation of assessment instruments commonly employs the techniques of content validity, face validity, criterion validity, and construct validity [28]. The validation of this instrument, named Work Environment (WE), tested content validity, construct validity, reliability, criterion validity, and sensitivity.

The first step to validate the instrument was testing content validity, seeking to identify the clarity of language, practical pertinence, and theoretical relevance. This verification was conducted by evaluator-judges selected according to their field, following the procedures described by the authors [29]: the instrument was forwarded to 12 evaluator-judges with doctorate degrees in the corresponding area of study. They assessed each of the 50 questions that composed the instrument according to their theoretical relevance. The evaluator-judges rated the questions on a five-point Likert scale, determining the degree of validity of each indicator. For the items that received a score lower than three, alteration suggestions were requested [29]. According to the evaluator-judges, the instrument presented good and excellent accordance regarding its content validity. No item received a score less than three.

After content validation, a pilot test was applied to a group of 100 workers of the company Beta, the focus of this study, aiming to test the feasibility of the study’s design and the clarity of the instrument. In this moment all workers signed an Informed Consent, of ethics committee project CAAE 35553720.0.0000.5547. The test revealed that the workers understood the instrument, eliminating the need for changes.

Subsequently, the construct validity was tested through exploratory and confirmatory factor analysis. To allow this analysis, the aforementioned company Beta, which employs 565 workers, was contacted and authorized the survey. For a reliability level of 95% and a margin of error of 5%, the necessary sample would be 229 workers. For its validation, the instrument was applied to 231 workers, belonging to the same group as the pilot study, at this moment, all workers signed the Free and Informed Consent Term, according to the Ethics Committee project CAAE 35553720.0.0000.5547. These company employees over 18 years old were selected, and those who would be excluded would be given incomplete or blank questionnaires; however, no questionnaire was given this way.

In order to assess construct validity, Pearson’s correlation was employed. Pearson’s correlation is commonly used to verify the intensity of the existing linear association between variables and it measures the linear association between quantitative variables [30]. This coefficient is a number between −1 and 1 that expresses the degree of linear dependence between two quantitative variables. If negative, it indicates that one variable decreases as the other increases; if positive, it indicates that one variable increases as the other increases [31,32,33,34]. The *r* values are distributed as follows: *r* = 0–0.25, very low correlation; *r* = 0.26–0.49, low correlation; *r* = 0.5–0.69, moderate correlation; *r* = 0.7–0.89, high or strong correlation; *r* = 0.9–1.0, very high or very strong correlation [35]. Pearson’s correlation was employed because the instrument presents linear association between the criteria presented.

The next step was reliability analysis through test–retest of the instrument and analyzing its internal consistency. Reliability or trustworthiness seeks to identify how consistent are the scores obtained by the same subjects surveyed when they are re-examined, which occurs through the application of the same instrument on another occasion [36].

The test–retest procedure was employed to verify the instrument’s reliability. The instrument was applied by the researcher in the same company Beta and the workers answered the questions (test). After seven days, it was reapplied to the same group (retest).

The instrument’s reliability was verified through Cronbach’s alpha, calculating the existing correlation between each item of the test and the remaining items or the total (total score). Cronbach’s alpha has particular characteristics that seek solutions in a population surveyed, not simply a measure by itself, and these alpha values change according to the group in which the measure is adapted [37].

Having verified the instrument’s reliability, the next step would be the criterion validation, aiming to examine whether the measuring scale was adequate to what was expected in relation to other variables, which are selected through expressive criteria [38,39]. For this instrument, it was not possible to validate this aspect, as there are no tools that assess the work environment in the same manner to compare with the proposed instrument. There are only individual instruments in the areas that support this instrument. Considering that, it was not possible to validate the criteria, which may be pointed as a limitation of the study.

Finally, a sensitivity analysis sought to ascertain whether the instrument was identifying its actual goal. The sensitivity analysis aimed to identify whether this study actually measured its intended objective among the population assessed and to detect significant changes in the construct studied [40].

The instrument’s sensitivity was assessed through indicators of central tendency and dispersion (mean standard deviation, maximum score, minimum score, and amplitude) of both sexes, only men, and only women so as to compare the results. These data were obtained in the application of the instrument during the test stage, and they revealed that the instrument measured its intended target.

The stages that required statistical assessment were performed through SPSS 22.0 (Statistical Package for Social Science, IBM, Armonk, NY, US) for Windows.

## 3. Results and Discussion

### 3.1. Anthropometric Characteristics

In order to validate this instrument, a survey was conducted with 231 collaborators of the aforementioned company, the focus of this study. The anthropometric data showed that the respondents were, on average, 32.3 years old (SD ± 10.91). The sample contained 139 male respondents and 92 female respondents. In terms of educational background, 9 had basic education, 70 had a high school degree, 42 were undergraduate students, 53 had a college degree, and 57 had post-graduate specializations. These 231 collaborators were distributed in the three hierarchical levels of the company (strategic, tactical, and operational). Within the questionnaires, information about the organization’s hierarchical levels was collected, but it did not include this as a demographic variable in order for this instrument to meet all hierarchical levels. Also pointed out that the hierarchical pattern was implicit in the educational pattern and that this varied from company to company.

Thus, the anthropometric data revealed that the respondents were 32 years old on average, mostly male, had a high school degree, and were mostly single.

### 3.2. Construct Validation

The Kaiser–Meyer–Olkin (KMO) index, used to verify the suitability of the application of Exploratory Factor Analysis (EFA) for this study’s data set, returned a value of KMO = 0.917, which demonstrated that the data were suitable for factor analysis. Bartlett’s sphericity test was significant (*χ²* = 7465.349, *Df* = 1225, *p* ≤ 0.000), allowing the EFA to be conducted.

Through the principal component method to estimate the factor loadings and specificity, the EFA was conducted. It adopted the varimax rotation method, in a correlation matrix composed of 50 variables (questions).

The number of factors estimated was determined by assessing the scree plot, employing question retention through the Kaiser–Guttman criterion of the components with eigenvalues higher than 1 [41]. The analysis showed that nine factors should be retained (1–9), as they presented eigenvalues higher than 1. However, they would not represent the instrument as a whole. For that reason, the EFA for one factor is described below.

Seeking to test the unidimensionality of the instrument, the EFA was conducted for one factor through commonality, employing the principal component method. Table 2 exhibits the question and commonality value for each of the instrument’s 50 questions.

Only question 30 presented commonality lower than 0.5. The others presented values higher than 0.5, demonstrating the existence power of each question. According to [42], the higher the commonality, the greater the explanatory power of a determined item in the question model, and each must present commonality higher than 0.5. However, question 30 is the only one that addressed the topic “autonomy granted by the company to make decisions”. Therefore, for prudence, the question was retained despite its commonality value of 0.456, inferior to the value indicated by [42].

After the commonality analysis employing the principal component method, the component matrix was verified. Employing the principal component analysis as the extraction method and the varimax rotation method with Kaiser normalization, the instrument went from 14 factors initially to 10 factors after the statistical analysis.

The 50 questions were distributed into 10 factors, each allocated in one or more domains. However, the extraction and rotation method revealed the highest score in the factors, determining in which one the question must remain. Table 3 displays this analysis.

It was possible to observe that factor 1 was composed by questions 22–25, 28–30, 39, and 43; factor 2 by questions 26, 27, 31–33, 45–47, and 50; factor 3 by questions 4, 8–11, and 18; factor 4 by questions 34–37, 41, and 42; factor 5 by questions 1–3, 5, and 6; factor 6 by questions 15–17; factor 7 by questions 19–21; factor 8 by questions 38, 40, 44, 48, and 49; factor 9 by questions 12–14; and factor 10 was composed by question 7.

The factors were named according to the area which they encompassed, as pointed by the instrument’s theoretical construct and displayed by Table 4.

Factor 1—Health comprised five questions (1–3, 5, and 6). Factor 2—Emotional and Psychological comprised six questions (4, 8–11, and 18). Factor 3—Spiritual comprised three questions (12–14). Factor 4—Sleep and Rest comprised only one question (7). Factor 5—Work and Life comprised three questions (15–17). Factor 6—Work Conditions comprised three questions (19–21). Factor 7—Leadership Management comprised nine questions (22–25, 28–30, 39, and 43). Factor 8—Remuneration and Functional Assistance also comprised nine questions (26, 27, 31–33, 45–47, and 50). Finally, Factor 10—Personal Relations and Company Image comprised five questions (38, 40, 44, 48, and 49).

Seeking to analyze the construct validity of the instrument, the correlation matrix demonstrated the existing relationships between the pairs of variables.

In the EFA application, Pearson’s correlation was conducted in the test–retest stage of the instrument, adopting a significance level of 0.01. The analysis of the correlation matrix revealed that all of the 50 questions presented a positive correlation.

According to [30], the *r* values from 0 to 0.25 indicate a very low correlation, from 0.26 to 0.49 indicate a low correlation, from 0.5 to 0.69 indicate a moderate correlation, from 0.7 to 0.89 indicate a high or strong correlation, and from 0.90 to 1.0 indicate a very high or very strong correlation. The results showed that none of the questions presented very low correlations. Eight questions presented low correlations, thirty-eight presented moderate correlations, and four presented high or strong correlations, as displayed in Table 5.

Thus, the 50 questions of the instrument were correlated. Even though eight questions presented a low correlation, they were significantly correlated.

### 3.3. Reliability

In order to determine the instrument’s internal consistency in terms of reliability, Cronbach’s alpha was calculated for the existing correlation between each item of the test and the remaining items or their total (total score). The Cronbach’s alpha of the 50 questions is exhibited in Table 6.

Although determinant values higher than 0.7 are considered ideal [43], values under 0.7, but close to 0.6, can be regarded as satisfactory [44].

The Cronbach’s alpha values presented in the instrument ranged 0.953–0.956. The only item that obtained 0.956 was question 5, while the others stayed within the interval mentioned.

The results showed that all of the factors presented values higher than 0.7, which indicated the reliability of the proposed instrument.

The full instrument is available in the Appendix A, with the 50 questions.

## 4. Conclusions

The construct validation was conducted through Exploratory Factor Analysis, obtaining the values for KMO (0.917) and Bartlett’s test (*χ²* = 7465,349, *Df* = 1225, *p* ≤ 0.000). The EFA composed of the total of variables in the instrument revealed that nine questions should be removed from the initial fourteen. For that purpose, an EFA was conducted for one question through the commonality analysis, which presented only one question under the necessary parameter. However, this question remained in the instrument because it was the only one that addressed the theme of autonomy granted by the company to make decisions.

After the commonality analysis, the component matrix was observed. The instrument, which at first featured fourteen factors, contained ten factors after the statistical analysis.

Seeking to analyze the construct validity of the instrument, the correlation matrix was examined in the application of the instrument’s test–retest. All of the questions of the matrix presented positive correlations. Eight questions presented low correlation, thirty-eight presented moderate correlation, and four presented high or strong correlation.

Regarding reliability, Cronbach’s alpha was calculated in order to determine the instrument’s internal consistency, pointing to values ranging from 0.953 and 0.956, which indicated that all of the questions presented reliability.

Thus, it is possible to conclude that the proposed instrument is valid and applicable, seeing that it met all of the parameters necessary for its validation. It assesses elements of QL, QWL, and OC as a single tool that allows managers to comprehend the situation of the work environment.

## Figures and Tables

**Table 1 ijerph-17-07364-t001:** Factors of the theoretical construct.

No.	FACTORS
**1**	Health
**2**	Emotional and Psychological
**3**	Spiritual
**4**	Work, Life, and Family
**5**	Work Conditions
**6**	Leadership
**7**	Fair Compensation
**8**	Professional Appreciation and Growth
**9**	Functional Assistance
**10**	Functional Capacity and Responsibility
**11**	Social Integration
**12**	Organizational Communication
**13**	Constitutionalism
**14**	Institutional Image

Source: the authors [26].

**Table 2 ijerph-17-07364-t002:** Exploratory question analysis for one question.

Question	Commonality	Question	Commonality
1	0.719	26	0.649
2	0.680	27	0.595
3	0.733	28	0.767
4	0.614	29	0.736
5	0.652	30	0.456
6	0.645	31	0.553
7	0.640	32	0.525
8	0.718	33	0.683
9	0.704	34	0.643
10	0.619	35	0.650
11	0.698	36	0.670
12	0.633	37	0.627
13	0.651	38	0.634
14	0.748	39	0.697
15	0.699	40	0.651
16	0.710	41	0.503
17	0.713	42	0.822
18	0.595	43	0.792
19	0.667	44	0.700
20	0.736	45	0.776
21	0.590	46	0.747
22	0.658	47	0.648
23	0.715	48	0.663
24	0.806	49	0.660
25	0.695	50	0.658

Extraction method: principal component analysis.

**Table 3 ijerph-17-07364-t003:** Principal component analysis.

Question	Factor									
	1	2	3	4	5	6	7	8	9	10
1					0.751					
2			0.391		0.643					
3					0.814					
4			0.613							0.334
5					0.740					
6					0.509					0.493
7										0.658
8			0.706							0.327
9			0.807							
10			0.714							
11			0.783							
12									0.713	
13			0.413						0.527	
14									0.734	
15						0.684				
16						0.703				
17						0.680				
18			0.456		0.342	0.371				
19							0.728			
20							0.783			
21							0.644			
22	0.420					0.391	0.303			0.320
23	0.740									
24	0.743			0.313						
25	0.715									
26	0.421	0.425				0.397				
27		0.694								
28	0.666	0.421								
29	0.681	0.331								
30	0.541									
31	0.406	0.460						0.302		
32		0.536		0.318						
33		0.690								
34				0.557		0.372				
35				0.645						
36				0.725						
37				0.632						
38	0.359							0.631		
39	0.533							0.439		
40		0.300						0.513		
41	0.326	0.682						0.318		
42	0.347	0.634						0.318		
43	0.683									
44	0.348			0.389				0.506		
45		0.566		0.447			0.390			
46		0.620		0.385						
47	0.303	0.463		0.338			0.319			
48		0.311						0.572		
49								0.532		
50		0.591		0.349						

Extraction method: principal component analysis; rotation method: varimax with Kaiser normalization.

**Table 4 ijerph-17-07364-t004:** Factor names according to the question distribution.

Factor	Question
1—Health	1–3, 5, 6
2—Emotional and Psychological	4, 8–11, 18
3—Spiritual	12–14
4—Sleep and Rest	7
5—Work and Life	15–17
6—Work Conditions	19–21
7—Leadership Management	22–25, 28–30, 39, 43
8—Remuneration and Functional Assistance	26, 27, 31–33, 45–47, 50
9—Functional Responsibility	34–37, 41,42
10—Personal Relations and Company Image	38, 40, 44, 48, 49

**Table 5 ijerph-17-07364-t005:** Classification of Pearson’s correlation.

Classification	Question *(r)*
Low *r* = 0.26–0.49	10 (*r* = 0.497)
	36 (*r* = 0.460)
	38 (*r* =0.305
	39 (*r* = 0.379)
	40 (*r* = 0.268)
	41 (*r* = 0.264)
	43 (*r* = 0.400)
	44 (*r* = 0.465)
Moderate *r* = 0.50–0.69	1 (*r* = 0.519)
	2 (*r* = 0.583)
	3 (*r* = 0.681)
	4 (*r* = 0.647)
	6 (*r* = 0.597)
	8 (*r* = 0.627)
	9 (*r* = 0.653)
	11 (*r* = 0.562)
	12 (*r* = 0.653)
	13 (*r* = 0.574)
	14 (*r* = 0.596)
	15 (*r* = 0.676)
	16 (*r* = 0.556)
	17 (*r* = 0.640)
	18 (*r* = 0.641)
	21 (*r* = 0.675)
	22 (*r* = 0.698)
	23 (*r* = 0.593)
	24 (*r* = 0.633)
	25 (*r* = 0.622)
	26 (*r* = 0.623)
	27 (*r* = 0.629)
	28 (*r* = 0.673)
	29 (*r* = 0.659)
	30 (*r* = 0.557)
	31 (*r* = 0.555)
	32 (*r* = 0.621)
	33 (*r* = 0.598)
	34 (*r* = 0.615)
	35 (*r* = 0.500)
	37 (*r* = 0.560)
	42 (*r* = 0.625)
	45 (*r* = 0.618)
	46 (*r* = 0.641)
	47 (*r* = 0.682)
	48 (*r* = 0.576)
	49 (*r* = 0.616)
	50 (*r* = 0.630)
High or strong *r* = 0.70–0.89	5 (*r* = 0.741)
	7 (*r* = 0.700)
	19 (*r* = 0.702)
	20 (*r* = 0.701)

**Table 6 ijerph-17-07364-t006:** Cronbach’s alpha, if the items were discarded *(α-x).*

Question	(*α-x*) *	Question	(*α-x*) *
1	0.954	26	0.953
2	0.954	27	0.953
3	0.955	28	0.953
4	0.954	29	0.953
5	0.956	30	0.954
6	0.954	31	0.953
7	0.955	32	0.954
8	0.954	33	0.953
9	0.954	34	0.954
10	0.954	35	0.954
11	0.954	36	0.954
12	0.954	37	0.954
13	0.954	38	0.953
14	0.954	39	0.954
15	0.953	40	0.953
16	0.953	41	0.954
17	0.953	42	0.953
18	0.954	43	0.953
19	0.954	44	0.953
20	0.954	45	0.953
21	0.954	46	0.953
22	0.953	47	0.953
23	0.953	48	0.953
24	0.953	49	0.953
25	0.953	50	0.953

* Cronbach’s alpha.

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
