# Peer review of "Construct Validity and Reliability of the Work Environment Assessment Instrument WE-10"

_ijerph, 2020, doi:10.3390/ijerph17207364_

Round 1

Reviewer 1 Report

First of all, thank you for the opportunity to review this interesting work, which in my opinion deserves to be published.

The purpose (aim) of study is, from an organizational research perspective, very relevant.  The research contributes to having a valid and reliable instrument to measure organizational aspects of the work environment. Likewise, the methodological process to validate the questionnaire is methodologically rigorous and well exposed. As a suggestion to increase the clarity of the exposition: I would change the term factor (by item) and the term domain (by factor).

On the other hand, some questions I think deserve to be clarified or considered.

- In the introduction, the theoretical development is very schematic. For example, the concepts that feed the questionnaire are poorly defined. Both the term QWL and QL have little explanation, but in particular, the concept of organizational climate (which is a term that can be applied in a broad and diverse way) I think deserves more attention.

- In the demographic variables of the sample, I do not think marital status relevant (being the instrument designed for the organizational study). Likewise, a variable related to organizational status (such as hierarchical levels or job categories) appears to be missing.

- Finally, have the authors considered the possibility of attaching the items of the questionnaire? Alternatively, I think it would be desirable to have some sample items (or factors) for each of the factors (domains).

Author Response

The purpose (aim) of study is, from an organizational research perspective, very relevant.  The research contributes to having a valid and reliable instrument to measure organizational aspects of the work environment. Likewise, the methodological process to validate the questionnaire is methodologically rigorous and well exposed. As a suggestion to increase the clarity of the exposition: I would change the term factor (by item) and the term domain (by factor).

The terms for question and factor were changed

On the other hand, some questions I think deserve to be clarified or considered.

- In the introduction, the theoretical development is very schematic. For example, the concepts that feed the questionnaire are poorly defined. Both the term QWL and QL have little explanation, but in particular, the concept of organizational climate (which is a term that can be applied in a broad and diverse way) I think deserves more attention.

Added a paragraph in line 67 addressing this relationship

- In the demographic variables of the sample, I do not think marital status relevant (being the instrument designed for the organizational study). Likewise, a variable related to organizational status (such as hierarchical levels or job categories) appears to be missing.

Marital status was removed and information on hierarchical levels was added

- Finally, have the authors considered the possibility of attaching the items of the questionnaire? Alternatively, I think it would be desirable to have some sample items (or factors) for each of the factors (domains

The instrument was developed in Portuguese, translated into English and attached to the article

Reviewer 2 Report

Article with sufficient quality to be published, after some minor revisions:

  • Further clarification of the rationale behind the definition of the theoretical construct model and initial selection of the different domains and factors;
  • Explanation of the methodology used by the 5 experts to assess the instrument and domains and factors selection;
  • Presentation of the number of factors identified by the 12 evaluator-judges 
    with a score lower than three in testing content validity, and changes made;
  • Explanation of the inclusion criteria and participants selection method in the different validation studies;
  • Brief characterization of participants in the different validation studies;

Author Response

-Further clarification of the rationale behind the definition of the theoretical construct model and initial selection of the different domains and factors;

Added a paragraph in line 67 addressing this relationship

-Explanation of the methodology used by the 5 experts to assess the instrument and domains and factors selection;

Added in line 100 of the experts, specialists in psychometric assessment of instruments and work in the areas covered by the instrument

-Presentation of the number of factors identified by the 12 evaluator-judges with a score lower than three in testing content validity, and changes made;

Added in line 127 an addendum to the article that no question fell below 3

-Explanation of the inclusion criteria and participants selection method in the different validation studies;

Add in line 137 company employees over 18, and those who would be excluded would be incomplete

-Brief characterization of participants in the different validation studies;

Added in the methodology that the same sample was used in all stages.

Reviewer 3 Report

The study is properly designed and performed.

Authors assumed that the work environment include quality of life (QL), quality of work life (QWL), and organizational climate (OC). Then they designed and validated the construct and reliability of an instrument to assess the work environment as a single tool based on these areas.

Analysis is clear so are the results. It is possible to conclude that the proposed instrument is valid and applicable, seeing that it met all of the parameters necessary for its validation. It assesses elements of QL, QWL, and OC as a single tool that allows the manager to comprehend the situation of the work environment. The colaborated managerial tool can occur useful for management practice, still being based on scientific foundations.

My restriction is some confusion of concepts. In factor analysis, FACTORS are referred to as some general "domains", which are independent latent variables. Ech of them is loaded by few "indicators" or "observable variables". In the text, Authors call these observable variables "factor", which is a methodological mistake. It should be changed in the whole text, that "questions" asked to the emplyees are "observable variables" and FACTORS (independent latent variables) are what the Authors called so far "domains".

Author Response

-My restriction is some confusion of concepts. In factor analysis, FACTORS are referred to as some general "domains", which are independent latent variables. Ech of them is loaded by few "indicators" or "observable variables". In the text, Authors call these observable variables "factor", which is a methodological mistake. It should be changed in the whole text, that "questions" asked to the emplyees are "observable variables" and FACTORS (independent latent variables) are what the Authors called so far "domains".

The terms for question and factor were changed OK
